# Prevalence and Antimicrobial Resistance of *Campylobacter jejuni* and *Campylobacter coli* in Wild Birds from a Wildlife Rescue Centre

**DOI:** 10.3390/ani12202889

**Published:** 2022-10-21

**Authors:** Gaia Casalino, Francesco D’Amico, Francesca Rita Dinardo, Giancarlo Bozzo, Valeria Napoletano, Antonio Camarda, Antonella Bove, Roberto Lombardi, Francesco Paolo D’Onghia, Elena Circella

**Affiliations:** 1Department of Veterinary Medicine, University of Bari “Aldo Moro”, S.P. Casamassima km. 3, 70010 Valenzano, Italy; 2Regional Wildlife Rescue Centre, Via Generale Palmiotti Michele, 43, 70020 Bitetto, Italy

**Keywords:** *Campylobacter* *jejuni*, *Campylobacter coli*, antimicrobial resistance, wild birds, wildlife rescue centre, biosecurity

## Abstract

**Simple Summary:**

Wildlife may be zoonotic carriers of pathogens. The aim of this paper was to investigate the presence of *Campylobacter (C.) jejuni* and *C. coli,* which are responsible for enteric syndromes and sometimes extraintestinal diseases in humans, in wild birds housed at a wildlife rescue centre. Sensitivity to antibiotics was also investigated in the detected strains. Two hundred and nine birds belonging to 33 different species were considered in the study. *Campylobacter* was found in 52 birds (24.88%), mainly in omnivorous species. In terms of housing conditions, the infection rate was higher in birds housed in indoor (57.14%) than in outdoor aviaries (31.74%). Interestingly, *Campylobacter* was not detected in some species whose mean temperature body was below 40 °C or higher than 42.2 °C. All detected strains were *C. jejuni* except for three *C. coli* that were identified in Long-eared Owls. The most commonly found antibiotic resistance was against drugs such as trimethoprim/sulfamethoxazole, ciprofloxacin and enrofloxacin. Multi-drug resistance against two or more antibiotics was also detected. The findings of the study highlight the relevance of increasing biosecurity measures at the wildlife rescue centres to reduce health risks to staff involved in wildlife management.

**Abstract:**

Climate change, excessive exploitation of agricultural land which reduces natural habitats, wildlife shooting, and the use of pesticides all cause difficulties for wildlife, with considerable numbers of animals being brought to wildlife rescue centres. Although the efforts of staff involved in wildlife management at these centres usually focus on therapeutic treatments to reintroduce them into the wild, the monitoring of pathogens that may be transmitted to humans is of relevance. *Campylobacter* (*C.*) *jejuni* and *C. coli* are frequently carried by animals without inducing clinical signs and are responsible for enteric disorders and more rarely extra-intestinal disease in humans. Farm species and poultry, in particular, are the main reservoirs of *C. jejuni* and *C. coli*, but wild animals may also be carriers. The aim of this paper was to investigate the presence of *C. jejuni* and *C. coli* in wild birds housed at a wildlife rescue centre and to evaluate the sensitivity of the detected strains to antibiotics. *Campylobacter* was found in 52 out of 209 (24.88%) birds from 33 different species. *C. jejuni* was more prevalent, while *C. coli* was only detected in three Long-eared Owls (*Asio otus*). The incidence of the infection was particularly high (72.22%) among omnivorous species. Infection rates were higher in birds housed indoors (57.14%) than outdoors (31.74%). Moreover, *Campylobacter* was not detected in species whose mean temperature body is below 40 °C or higher than 42.2 °C. The most common antibiotic resistance in the tested strains was against trimethoprim/sulfamethoxazole, ciprofloxacin and enrofloxacin. In addition, multi-drug resistance was also found. The results highlight the need to increase biosecurity measures at rescue centres so as to reduce health-related risks to workers involved in wildlife management.

## 1. Introduction

The impact of human activities on ecosystems is often negative, amongst other things inducing imbalances between animal species that coexist in natural environments. Some of the most worrying effects deriving from human activities are pollution and climate change at a global level [1,2]. In summer, for example, environmental temperatures of above 40 °C have often persisted for many days or weeks in several areas, with negative effects on wildlife, particularly on nestlings or young animals which are more prone to dehydration. In addition, shooting, especially where improperly practised, puts pressure on wild animal populations in nature [3,4]. Overexploitation of agricultural lands, often used for monocultures, has led to the alteration of natural habitats. Moreover, pesticide use in agriculture leads to a reduction in trophic availability for insectivorous species as well as to possible toxic events [5]. All these jeopardise wildlife, with many animals needing to be rescued and taken to wildlife recovery centres. At these centres, it is important to treat the animals in order to reintroduce them into the wild, but also to monitor any infections that might otherwise be transmitted to humans, so as to protect staff engaged in rehabilitation. There are many pathogens that can be transmitted by animals, and, of those, *Campylobacter*, which belongs to the *Campylobacteraceae* family, can affect human health. It is asporogenous, Gram-negative, mobile, microaerophilic, with a spiral shape and size ranging from 0.2 to 0.9 micrometres. It is thermotolerant, with its growth temperature ranging between 37 °C and 42 °C. Within the genus *Campylobacter*, the best-known species are *Campylobacter jejuni* and *Campylobacter coli*, as they are linked with campylobacteriosis in humans [6].

In humans, *Campylobacter* is responsible for abdominal pain, profuse diarrhoea, nausea, and fever, that usually resolves within 5–8 days [7,8]. Children aged 1–4 years old, 15–24-year-olds and seniors are particularly sensitive to the effects of *Campylobacter*. Complications such as bacteraemia, meningitis, pancreatitis, cholecystitis, nephritis, myocarditis, and hepatitis can also occur [9], particularly in immunocompromised patients [6]. A possible sequela of campylobacteriosis associated with *C. jejuni* in humans is Guillain-Barré syndrome, a serious autoimmune-mediated neurological disorder. Although the syndrome can be linked to other possible causes, about 30% of cases have been attributed to a previous *C. jejuni* infection [10]. Guillain-Barré syndrome is characterised by progressive paralysis of the limbs, weakness, and decreased sensitivity to pain. A rare variant of Guillain-Barré syndrome is Fisher Miller syndrome which is linked to *C. jejuni* [11,12], although not exclusively [12]. Ataxia, areflexia and bilateral ophthalmoplegia are typical of Fisher Miller syndrome while dysesthesia of the limbs, facial, bulbar, and papillary paralysis occur less frequently. Another sequela of *C. jejuni* infection in humans is Reiter’s syndrome, an autoimmune spondyloarthropathy which mainly affects individuals aged between 15 and 35 years [9]. Moreover, both in humans and animals including cattle (*Bos taurus*), sheep (*Ovis aries)*, goats (*Capra hircus*) and pigs (*Sus scrofa*), *Campylobacter* can cause genital infections. *C. jejuni*, *C. coli*, *C. fetus* and *C. upsaliensis* can lead to neonatal meningitis or foetal death in pregnant animals [9].

Campylobacter is a food-borne pathogen prevalently linked to poultry meat; indeed, poultry is a major source of infection for humans. Nevertheless, other risk factors include foreign travel, especially to developing countries [13], drinking contaminated water or raw milk, eating unpasteurized cheeses or undercooked sheep, pig, and shellfish meat, and contact with pets [6]. Although poultry is one of the most important reservoirs of *Campylobacter* [14], pet [15] and wild birds can also be carriers for the bacterium [16,17]. Pigeons, which come into contact with humans in urban centres, can be carriers of *C. jejuni* and other pathogens such as *Salmonella* spp. and *Chlamydia psittaci*. The rate of positivity for *C. jejuni* infection can reach 26.2%, particularly in areas with high densities of individuals [18]. Another study highlighted that domestic and feral pigeons carried *C. jejuni* and *C. coli* [19]. In New Zealand, where campylobacteriosis occurs especially in children, local wildlife and pets have been identified as one of the potential risk factors for the infection [20]. *C. jejuni* has been found in several species, mainly in ducks and dogs in summer, and in starlings in winter. Otherwise, it has been found in swans, geese and wild cats without seasonal prevalence [20]. In Europe, where campylobacteriosis is considered one of the most worrying zoonoses, farm species such as cattle, pigs and poultry (*Gallus gallus*) are considered the main reservoirs of the pathogen, but wild species that choose areas adjacent to farms for breeding and feeding also seem to contribute to the spread of *Campylobacter* among domestic species. Wild species such as the common blackbird (*Turdus merula*) and sparrows (*Passer domesticus*, *Passer montanus*) that feed on the ground may play a more important role in spreading the bacterium compared to species that hunt in flight [21]. Furthermore, rodents such as mice and rats, that can reproduce very quickly, are vectors of several pathogens. Among wild rodents, the bank vole (*Myodes glareolus*) can host *C. jejuni* [22]. *C. jejuni* has been detected in faecal samples from wild common raccoons (*Procyon lotor*) and masked palm civets (*Paguma larvata*) in Japan [23]. Among the cervids, *C. jejuni* has been found in roe deer [24]. *Campylobacter jejuni, C. coli* and *C. lanianae* have been detected in wild boars [25]. *C. jejuni* has also been found in red squirrels in southern Italy [26]. *Campylobacter* spp. has also been identified in the oral cavity, from dental plaque, and on the gastric mucosa of dolphins (*Tursiops gephyreus*) [27]. *Campylobacter lari* subsp. *lari* has been detected in penguins belonging to *Pygoscelis adeliae* and *Pygoscelis papua,* and *Campylobacter lari* subsp. *concheus* in *Pygoscelis adeliae* and *Pygoscelis antarcticus* [28]. *Campylobacter* has also been found in wild birds living in the wild and housed in rescue centres [16,17,29,30].

The aims of this study were to (i) investigate the occurrence of *Campylobacter* among wild birds housed in a wildlife rescue centre, (ii) assess the most frequently carried *Campylobacter* species, and (iii) evaluate the sensitivity to antibiotics of the detected strains.

## 2. Materials and Methods

### 2.1. Examined Birds and Collected Samples

Two hundred and nine wild birds housed at the Regional Wildlife Rescue Centre (RWRC), of Apulia, Bitetto, BA, Italy, were tested for *Campylobacter*. Those birds had previously been hospitalised at the RWRC due to debilitation, dehydration and traumatic lesions, or else arrived as nestlings or unweaned birds, incapable of surviving on their own in the wild. The birds were from 33 different species (Table 1) and were housed in either indoor or outdoor aviaries.

A cloacal swab sample was collected from each bird. In addition, the body temperature of the individuals of some species was measured with a thermometer. Bird-handling and sample collections were performed according to the guidelines of the Ethics Committee for Animal Experimentation of the Department of Veterinary Medicine (DiMeV), Italy.

### 2.2. Identification of Campylobacter

All cloacal samples were put into sterile tubes containing 5 mL *Campylobacter* selective enrichment broth (OXOID, Basingstoke, UK), previously enriched with 5% of sheep blood, *Campylobacter* Selective Supplement (SR0085E) (OXOID) and *Campylobacter* Growth Supplement (SR0232) (OXOID). Samples were incubated at 42 °C in microaerophilic conditions for 24 h. Next, they were plated into Petri dishes containing *Campylobacter* agar base (CM0689) (OXOID) previously sheep blood (5%) *Campylobacter* Selective Supplement (SR0098E) (OXOID) and *Campylobacter* Growth Supplement (SR0232) (OXOID). All Petri dishes were incubated at 42 °C in microaerophilic conditions for 48–72 h. The colonies morphologically compatible with *Campylobacter* spp. were individually transferred onto blood agar (Tryptic Soy Agar enriched with 5% sheep blood), to which *Campylobacter* Selective Supplement (SR0085E) (OXOID) was added.

A single colony from each sample was chosen and dissolved in 100 µL of distilled water. DNA extraction was obtained by heat treatment at 95 °C for 10 min using a thermal cycler (Mastercycler EP-Gradient, Eppendorf AG 22331, Hamburg, Germany). Multiplex-Polymerase Chain Reactions (PCR) were set up according to a previously described protocol [33] with minor modifications, obtaining the identification of *Campylobacter* genus and *C. jejuni* and *C. coli* species (Table 2).

The reaction mixture consisted of: iTaq buffer 10×, MgCl_2_ 50 mM, dNTPs 10 mM (of each of the four oligonucleotides), 11 μM of MD16S1 and MD16S2 and 10.42 μM of the remaining two pairs of primers, 1.34 U of iTaq DNA polymerase Platinum II Green HS PCR MM (Invitrogen, Vilnius, Lithuania), 2 μL of sample DNA, sterile distilled water to a final volume of 25 μL.

Cycling conditions were as follows: 94 °C for 5 min for 1 cycle; 94 °C for 15 s, 60 °C for 15 s, 72 °C for 10 s for 34 cycles; and 72 °C for 10 min for the final elongation. Two µL of Multiplex-PCR products were visualized by agarose gel electrophoresis (1.5%) which was stained with ethidium bromide (0.5 μg/mL), using TBE (Tris-Borato-EDTA, AppliChem, Ottoweg, 4, Darmstadt, Germany) as conductor. Ez PCR Molecular Ruler 100bp (Bio-Rad, Hercules, CA, USA) was used as reference marker. The results were read with Gel Doc-It (UVP, Upland, USA) image analyzer.

### 2.3. Antibiotic Susceptibility Testing

Fifty-one strains were tested to determine the susceptibility to azithromycin (AZM) 15 µg; chloramphenicol (CHL) 30 µg; ciprofloxacin (CIP) 5 µg; enrofloxacin (ENR) 5 µg; erythromycin (E) 15 µg; gentamicin (CN) 10 µg; nalidixic acid (NA) 30 µg; tetracycline (TE) 30 µg; and trimethoprim-sulfamethoxazole (SXT) 25 µg. Antibiotic susceptibility tests were performed on Muller–Hinton agar (OXOID, Basingstoke, UK) supplemented with 5% horse blood using the standard Kirby–Bauer disk diffusion method according to the European Committee on Antimicrobial Susceptibility Testing [34,35].

### 2.4. Statistical Analysis

The wild bird data were analyzed by univariate statistical analysis (Pearson’s chi-square test and Fisher’s exact test for independence) using *Campylobacter* spp. status (positive/negative) as the dependent variable. The odds ratio (OR) and 95% confidence interval (CI_95%_) were also calculated. Values of *p* < 0.05 were considered statistically significant. Statistical analyses were performed using Statistical analysis was performed using spss 13 software for Windows (SPSS Inc., Chicago, IL, USA).

## 3. Results

### 3.1. Detection of Campylobacter jejuni and C. coli

*Campylobacter* was identified in 52 out of 209 wild birds (24.88%) (Table 3 and Table 4). *C. jejuni* was found in 49 individuals (94.23%) and was thus prevalent with respect to *C. coli* that was identified only in 3 birds (5.77%), all of which were long-eared owls. *Campylobacter* was particularly found in some groups such as species belonging to *Corvidae* family (72.22%; *p* < 0.001; OR: 20.37, CI_95%_: 5.35–77.50) while it was not identified in others such as granivorous and freshwater species. Among the birds of prey, the rates of *Campylobacter* infection in nocturnal species (36.67%; *p* < 0.001; OR: 4.54, CI_95%_:1.67–12.31) was significantly higher compared to diurnal birds (11.32%). The kestrel was the only species that tested positive among the group of diurnal birds of prey. *Campylobacter* was found in 6 out of 23 kestrels while the individuals belonging to the other species tested negative. Among the nocturnal species, the long-eared was the owl most frequently carrying the infection, with 14 out of 31 birds testing positive (45.16%). *Campylobacter* was found in 6 out of 17 little owls (35.29%) and in 2 out 5 (40%) scops owls, while barn owls were all negative.

Despite not being statistically significant (*p* > 0.05), *Campylobacter* was prevalently detected in birds reared indoors (57.14%) rather than outdoors (31.74%) (Table 5).

Considering the species in which a correlation between the presence of *Campylobacter* and the age of the specimen could be made, the rate of infection was similar both in young and adult long-eared owls, while it was higher in nestlings/young (42.86%) than in adult kestrels (18.75%) (Table 6). Nevertheless, when comparing the two macro-categories nestlings/sub-adults versus adults, no statistically significant prevalence emerged (Table 6).

### 3.2. Relationship between the Prevalence of C. jejuni and C. coli and the Body Temperature of Birds

On further investigations into the link between body temperature and susceptibility to infection, *Campylobacter* was found in 45.16% of long-eared owls, 35.29% of little owls, 26.09% of common kestrels, and 16.66% of lesser kestrels, whose average body temperatures ranged from 40.7 to 41.8 °C (Table 7). For these species, the average body temperature of positive individuals was between 40.7 and 41.6 °C, whereas for those testing negative it fell into a range from 40.8 to 42.1 °C. *Campylobacter* was not found in barn owls, common buzzards and other large diurnal birds of prey, whose average body temperatures were 39.4, 42.5 and 42.2 °C, respectively. Comparing the susceptibility of these species to infection, a significantly higher *Campylobacter* rate was detected only in long-eared owl (*p* < 0.05; OR: 4.12, CI_95%_:1.25–13.57).

### 3.3. Antibiotic Resistance of C. jejuni and C. coli Strains

Among the *C. jejuni* strains, 25 (52.1%), 21 (43.7%) and 15 (31.2%) were resistant to trimethoprim/sulfamethoxazole, ciprofloxacin and enrofloxacin, respectively (Table 8). Ten strains (20.8%) were resistant to nalidixic acid and 6 (12.5%) to tetracycline. Moreover, one and two strains of *C. jejuni* were resistant to azithromycin and erythromycin, respectively. Likewise, among the tested *C. coli* strains, resistance was found for the same molecules except for erythromycin. No resistance was detected against chloramphenicol and gentamicin.

Multidrug resistance was found variously according to the different strains (Table 9). Resistance to three molecules was the most frequent multidrug resistance (21.56%) and CIP/ENR/SXT was the most detected association of molecules ineffective against the tested strains.

## 4. Discussion

*Campylobacter* was found in 52 out of 209 (24.88%) wild birds housed at the rescue centre. *C. jejuni*, the species most closely linked to human campylobacteriosis [6] was the most frequently identified *Campylobacter* species. This finding is of relevance considering that the infected birds were asymptomatic carriers that could more easily transfer the pathogen to the staff involved in the management of wildlife. *C. coli,* which is also linked to human campylobacteriosis, was more rarely detected, and was only identified in long-eared owls. Accordingly, *C. jejuni* was prevalent compared to *C. coli* and other *Campylobacter* species in wild birds from other wildlife rescue centres in Italy [36,37] and Spain [17], and in birds in the wild from Denmark [21]. According to the multiplex-PCR used in our study, which can detect species of *Campylobacter* other than *C. jejuni* and *C. coli* by way of genus primers, no other species of *Campylobacter* were identified in the tested birds. Likewise, no other species were detected in birds from wild rescue centres in Italy [36,37], while *Campylobacter lari* and a different, though unidentified, *Campylobacter* species were found in Spain, respectively in a long-eared owl (*Asio otus*) and in a tawny owl (*Strix aluco*) [17].

In our study, a different distribution of the incidence of *C. jejuni* infection based on bird species was observed among the wild birds housed in the rescue centre. Only long-eared owls sometimes harboured *C. coli*. A particularly high rate of infection (72.22%) was detected in omnivorous species, while *Campylobacter* was not found in parrots which are granivorous and frugivorous, although the number of parrots was very limited. Among raptors and owls, Strigiformes were more susceptible to infection (36.67%) than Accipitriformes (11.32%). These findings contrast with other studies involving birds of prey from wildlife recovery centres in central and southern Italy, which highlighted higher prevalence in raptors compared with owls, with positivity rates of 36.9% and 39.1% in raptors and 13.9% and 18.6% owls being found [36,37]. In raptors housed in a wildlife rescue centre in Spain, the rate of *Campylobacter* infection was lower (7.4%), with a similar distribution in diurnal (7.22%) and nocturnal species (7.89%) [17]. Although it could be related to a different epidemiological situation, the lower prevalence found in Spain could also be due to the different laboratory method used to detect *Campylobacter*. In fact, in the Spanish study, the samples were directly plated onto selective solid media, whereas enrichment with a selective nutrient broth before solid media is usually performed in screening procedures for the isolation of *Campylobacter* [30,38,39,40].

Adult wild birds usually have more opportunities to come into contact with *Campylobacter* in their lifetime, through several potential sources of environmental contamination. Our study found no statistically significant association between age and susceptibility to infection, though it did highlight a greater prevalence of *Campylobacter* infection in young kestrels compared to adult ones. A recent investigation was carried out on nestlings of Bonelli’s eagle (*Hieraaetus fasciatus*) in the wild [29]. Bonelli’s eagle is a bird of prey whose stronghold lies in the Iberian Peninsula which makes up about 65% of the European population. Cloacal swab and stool samples were collected from 45 nestlings. *Campylobacter* was identified in 4.7% of cloacal swabs, but not in any faecal samples, probably due to the poor survival of *Campylobacter* in the environment, confirming the importance of the type of sample when seeking to detect the bacteria [29]. The positivity of nestlings was very probably linked to the adults which may acquire the infection through their prey, mainly the European wild rabbit (*Oryctolagus cuniculus*) and the red partridge (*Alectoris rufa*). Where their usual prey species are lacking, particularly due to shooting, Bonelli’s eagles supplement their diet with pigeons, which are more likely to carry *Campylobacter* in their gut [18].

At the rescue centre, the higher rate of positivity detected in young individuals may well be linked to cross-contamination, caused by frequent contact with each other in the nursery. Likewise, although wild birds arriving at rescue centres may already be infected, birds housed in outdoor aviaries may be exposed to *Campylobacter* via free-living rodents or birds. Although not statistically significant, the higher rate of infection detected in birds housed indoors, particularly in long-eared owls, compared with individuals reared outdoors, could be linked to the closer contact between animals that can occur in a more confined environment. This finding confirms the need to increase biosecurity measures for birds and to avoid placing too many animals in the same facility.

The influence of diet on the incidence of *Campylobacter* infection in captive wild birds is lower than in the wild, particularly in the case of individuals housed in captivity for long periods. Indeed, even though the nutritional needs of each species are respected, the diet given to different species is more uniform at wildlife rescue centres than in natural environments. Nevertheless, a possible link between feeding behaviour and infection of birds should be considered. Diurnal raptors are fed mainly on birds, small mammals, and reptiles [41,42,43]. Owl diets mostly consist of mice, rats, and voles [44,45]. The long-eared owls which frequently tested positive for *Campylobacter* in this study have a wide spectrum of prey that also include birds [45], which amplify the species that are potential carriers of *Campylobacter.* Similarly, little owls tend to prey on mice, birds, reptiles, amphibians, and insects [46,47]. The barn owl diet is more strictly linked to vole, mole, mouse, and rat. Nevertheless, rodents can be carriers of *Campylobacter*, especially if they live in urban areas and feed on waste from human activity [48]. In our study, the fact that the barn owls tested negative could be due to unreleasable specimens at the wildlife centre being prevalently fed for prolonged periods with defrosted prey animals. In the case of scops owls, whose diet consists mainly of insects, grasshoppers, beetles and cicadas, invertebrates, earthworms, and spiders which are not linked with *Campylobacter*, two out five tested positive. Nevertheless, it should be considered that small amphibians, and micro-mammals [44,49] are also predated by scops owls. Additionally, possible indirect contact with positive individuals from other species at the rescue centre should not be excluded.

Gulls are also potential carriers of *Campylobacter* spp. with prevalence varying by species, age, and feeding habits [30]. Prevalence is also affected by the consumption of urban waste material abandoned along the coastline. In addition, *Campylobacter* in gulls may also be linked to the high content of urea, which is widely used as a substrate by the urease enzyme of *Campylobacter,* in gull excreta [8]. Among the gull species considered in our study, *Campylobacter* was only found in common gulls while the yellow-legged and Mediterranean gulls, whose diet is more strictly linked to fish, tested negative. Accordingly, a study carried out on the coasts of the north-eastern Iberian Peninsula and in the Medes Islands [50], where seagulls usually feed in landfills highlighted a high incidence of *Campylobacter* infection [50], while on the Columbretes islands, where gulls’ diets are made up almost exclusively of fish, the incidence was very low [50]. Positive gulls as well as other wild species could be responsible for spreading the bacterium on surface waters and drinking water reservoirs [51], with potential risks for human considering the possibilities of sharing the environments. Moreover, *Campylobacter* can be introduced into marine environments through sewage, leading some gulls to carry both *C. jejuni* and *C. lari*, as reported in Northern Ireland [8]. The contamination of surface water is higher in colder seasons compared to summer [51], when *Campylobacter* is probably reduced by the greater exposure to sunshine and ultraviolet radiation [52].

*Campylobacter* is a thermotolerant bacterium, with 42 °C being the temperature used for its isolation in vitro [53,54]. Poultry, whose body temperatures range from 41 to 42 °C [55], are the most common zoonotic vectors of *Campylobacter*. Among the wild species considered in this study, infection was particularly common in long-eared owls, little owls and kestrels, whose average body temperatures were 41.8, 41.5 °C and 41.4 °C, respectively. Interestingly, of these, individuals carrying *Campylobacter* had body temperatures averaging between 41.1 and 41.6 °C while those testing negative ranged from 41.8 to 42.1 °C. Comparing these species, a statistically significantly higher *Campylobacter* rate was detected only in long-eared owl; indeed, a trend seems to be emerging whereby the optimal temperature in vivo for *Campylobacter* is about 41.5 °C. Accordingly, barn owls, common buzzards, and larger birds of prey, whose average body temperatures were lower than 40 °C or higher than 42.2 °C all tested negative. Therefore, although the prevalence of *Campylobacter* infection could be affected by several factors, a relationship with body temperature should not be excluded. Considering that this is the first study looking into the possibility of such a relationship in wild birds, further investigations will be needed to address this issue.

With regard to antimicrobial resistance, all tested *Campylobacter* strains were susceptible to chloramphenicol and gentamicin. Only one strain of *C. jejuni* was resistant to erythromycin, but several strains resistant to drugs such as quinolones (ciprofloxacin, enrofloxacin, and nalidixic acid) and trimethoprim/sulfamethoxazole were found. These findings are relevant, considering that fluoroquinolones are the antibiotics most frequently used in veterinary medicine. Moreover, they are considered as valid alternative treatments to macrolides to treat *Campylobacter* infection in humans [56]. In addition, trimethoprim/sulfamethoxazole is the drug association most frequently used to treat enteric syndromes in humans. The resistance to azithromycin, a molecule only recently introduced into human medicine, was also found both in *C. jejuni* and *C. coli* strains. Moreover, multidrug resistance was found in some strains.

## 5. Conclusions

The incidence of *C. jejuni* and *C. coli* infection found in wild birds at the rescue centre highlights a potential risk for staff involved in wildlife management and the need to increase hygiene and biosecurity measures. Infection usually occurs in birds without clinical signs, thus raising the risk of infection in humans during the handling of wild birds. Moreover, the drug resistance found in the tested strains, as well as being a potential public health problem, also highlights another issue. Indeed, considering that antibiotic resistance can be transmitted from one bacterial species to another, it could lead to difficulties when treating sick and injured animals hospitalised at rescue centres.

## Figures and Tables

**Table 1 animals-12-02889-t001:** Species and number of specimens tested in the study.

	Family	Species	N°	Age and Housing (Outd: outdoor; Ind: indoor)			
Diurnal birds of prey	Falconidae	Eurasian Hobby (*Falco subbuteo*)	1	Nestling/young: 1		Adult: 0	
				Outd: 1	Ind: 0	Outd: 0	Ind: 0
		Common Kestrel (*Falco tinnunculus*)	23	Nestling/young: 7		Adult: 16	
				Outd: 7	Ind: 0	Outd: 16	Ind: 0
	Accipitridae	Eurasian Sparrowhawk (*Accipiter nisus*)	1	Nestling/young: 0		Adult: 1	
				Outd: 0	Ind: 0	Outd: 1	Ind: 0
		Peregrine Falcon (*Falco peregrinus*)	3	Nestling/young: 0		Adult: 3	
				Outd: 0	Ind: 0	Outd: 3	Ind: 0
		Red Kite (*Milvus milvus*)	3	Nestling/young: 0		Adult: 3	
				Outd: 0	Ind: 0	Outd: 3	Ind: 0
		Common Buzzard (*Buteo buteo*)	17	Nestling/young: 0		Adult: 17	
				Outd: 0	Ind: 0	Outd: 17	Ind: 0
		Western Marsh Harrier (*Circus aeruginosus)*	1	Nestling/young: 0		Adult: 1	
				Outd: 0	Ind: 0	Outd: 0	Ind: 1
		Short-toed Snake Eagle (*Circaetus gallicus*)	3	Nestling/young: 0		Adult: 3	
				Outd: 0	Ind: 0	Outd: 3	Ind: 0
		Pallid Harrier (*Circus macrourus*)	1	Nestling/young: 0		Adult: 1	
				Outd: 0	Ind: 0	Outd: 1	Ind: 0
Nocturnal birds of prey	Strigidae	Little Owl (*Athene noctua)*	17	Nestling/young: 0		Adult: 17	
				Outd: 0	Ind: 0	Outd: 17	Ind: 0
		Long-eared Owl (*Asio otus*)	31	Nestling/young: 25		Adult: 6	
				Outd: 21	Ind: 4	Outd: 6	Ind: 0
		Scops Owl (*Otus scops*)	5	Nestling/young: 0		Adult: 5	
				Outd: 0	Ind: 0	Outd: 3	Ind: 2
	Titonidae	Barn Owl (*Tyto alba*)	7	Nestling/young: 0		Adult: 7	
				Outd: 0	Ind:0	Outd: 7	Ind: 0
Strictly or prevalently insectivorous	Apodidae	Common Swift (*Apus apus*)	18	Nestling/young: 18		Adult: 0	
				Outd: 0	Ind: 18	Outd: 0	Ind: 0
	Falconidae	Lesser Kestrel (*Falco naumanni*)	30	Nestling/young: 0		Adult: 30	
				Outd: 0	Ind: 0	Outd: 28	Ind: 2
	Turdidae	Mistle Thrush (*Turdus viscivorus*)	2	Nestling/young: 0		Adult: 2	
				Outd: 0	Ind: 0	Outd: 2	Ind: 0
Omnivorous [31,32]	Corvidae	Eurasian Magpie (*Pica pica*)	10	Nestling/young: 1		Adult: 9	
				Outd: 0	Ind: 1	Outd: 4	Ind: 5
		Eurasian Jay (*Garrulus glandarius*)	5	Nestling/young: 1		Adult: 4	
				Outd: 1	Ind:0	Outd: 4	Ind: 0
		Western Jackdaw (*Coloeus monedula*)	1	Nestling/young: 0		Adult: 1	
				Outd: 0	Ind: 0	Outd: 1	Ind: 0
		Hooded Crow (*Corvus cornix*)	2	Nestling/young: 0		Adult: 2	
				Outd: 0	Ind: 0	Outd: 2	Ind: 0
Prevalently piscivorous	Laridae	Yellow-legged Gull (*Larus michahellis*)	1	Nestling/young: 0		Adult: 1	
				Outd: 0	Ind: 0	Outd: 1	Ind: 0
		Common Gull (*Chroicocephalus ridibundus*)	4	Nestling/young: 0		Adult: 4	
				Outd: 0	Ind: 0	Outd: 3	Ind: 1
		Mediterranean Gull (*Ichthyaetus melanocephalus)*	2	Nestling/young: 0		Adult: 2	
				Outd: 0	Ind: 0	Outd: 2	Ind: 0
	Ardeidae	Grey or Purple Heron (*Ardea cinerea, Ardea purpurea)*	2	Nestling/young: 0		Adult: 2	
				Outd: 0	Ind: 0	Outd: 2	Ind: 0
		Little Egret (*Egretta garzetta*)	1	Nestling/young: 0		Adult: 1	
				Outd: 0	Ind: 0	Outd: 1	Ind: 0
		Black-crowned Night Heron (*Nycticorax nycticorax*)	1	Nestling/young: 0		Adult: 1	
				Outd: 0	Ind: 0	Outd: 1	Ind: 0
	Phoenicopteridae	Greater Flamingo (*Phoenicopterus phoenicopterus*)	2	Nestling/young: 0		Adult: 2	
				Outd: 0	Ind: 0	Outd: 2	Ind: 0
Granivorous/ Frugivorous	Psittacidae	Monk Parakeet (*Myiopsitta monachus*)	4	Nestling/young: 4		Adult: 0	
				Outd: 4	Ind: 0	Outd: 0	Ind: 0
Aquatic birds	Anatidae	Common Shelduck (*Tadorna tadorna*)	1	Nestling/young: 0		Adult: 1	
				Outd: 0	Ind: 0	Outd: 1	Ind: 0
		Brent Goose (*Branta bernicla*)	2	Nestling/young: 0		Adult: 2	
				Outd: 0	Ind: 0	Outd: 2	Ind: 0
		Eurasian Teal (*Anas crecca*)	3	Nestling/young: 0		Adult: 3	
				Outd: 0	Ind: 0	Outd: 3	Ind: 0
		Mute Swan (*Cygnus olor*)	4	Nestling/young: 0		Adult: 4	
				Outd: 0	Ind: 0	Outd: 4	Ind: 0
		Mallard (*Anas platyrhynchos*)	1	Nestling/young: 0		Adult: 1	
				Outd: 0	Ind: 0	Outd: 1	Ind: 0

**Table 2 animals-12-02889-t002:** Primers and sequences used for the identification of *C. jejuni* and *C. coli*.

	Target Gene	Primers	Sequences	Amplicon Molecular Weight
Genus *Campylobacter*	*16S rRNA*	MD16 S1 MD16 S2	5′ATCTAATGGCTTAACCATTAAAC3′ 5′GGAGGGTAACTAGTTTAGTATT3’	857 bp
*C. jejuni*	*MapA*	MD mapA1 MD mapA2	5′CTATTTTATTTTTGAGTGCTTGTG3′ 5′GCTTTATTTGCCATTTGTTTTATTA3′	598 bp
*C. coli*	*CeuE*	COL3 MDCOL2	5′AATTGAAAATTGCTCCAACTATG3′ 5′TGATTTTATTATTTGTAGCAGCG3′	462 bp

**Table 3 animals-12-02889-t003:** Detection of *Campylobacter* in wild birds housed in the rescue centre.

	Family	Species	N° pos/ N° Tested	%
Diurnal birds of prey	Falconidae	Eurasian Hobby (*Falco subbuteo)*	0/1	0
		Common Kestrel (*Falco tinnunculus*)	6/23	26.09
	Accipitridae	Eurasian Sparrowhawk (*Accipiter nisus*)	0/1	0
		Peregrine Falcon (*Falco peregrinus*)	0/3	0
		Red Kite (*Milvus milvus*)	0/3	0
		Common Buzzard (*Buteo buteo*)	0/17	0
		Marsh Harrier *(**Circus aeruginosus)*	0/1	0
		Short-toed Snake Eagle (*Circaetus gallicus*)	0/3	0
		Pallid Harrier (*Circus macrourus*)	0/1	0
Nocturnal birds of prey	Strigidae	Little Owl (*Athene noctua)*	6/17	35.29
		Long-eared Owl (*Asio otus*)	14/31	45.16
		Scops Owl (*Otus scops*)	2/5	40.00
	Tytonidae	Barn Owl (*Tyto alba*)	0/7	0
Strictly or prevalently insectivorous birds	Apodidae	Common Swift (*Apus apus*)	2/18	11.11
	Falconidae	Lesser Kestrel (*Falco naumanni*)	5/30	16.66
	Turdidae	Mistle Thrush (*Turdus viscivorus*)	1/2	50.00
Omnivorous birds	Corvidae	Eurasian Magpie (*Pica pica*)	6/10	60.00
		Eurasian Jay (*Garrulus glandarius*)	5/5	100
		Western Jackdaw (*Coloeus monedula*)	1/1	100
		Hooded Crow (*Corvus cornix*)	½	50.00
Prevalently piscivorous birds	Laridae	Yellow-legged Gull (*Larus michahellis*)	0/1	0
		Common Gull (*Chroicocephalus ridibundus*)	2/4	50.00
		Mediterranean Gull (*Ichthyaetus melanocephalus*)	0/2	0
	Ardeidae	Grey or Purple Heron (*Ardea cinerea, Ardea purpurea*)	0/2	0
		Little Egret (*Egretta garzetta*)	0/1	0
		Black-crowned Night Heron (*Nycticorax nycticorax)*	0/1	0
	Phoenicopteridae	Greater Flamingo (*Phoenicopterus phoenicopterus*)	1/2	50.00
Granivorous and frugivorous birds	Psittacidae	Monk Parakeets (*Myiopsitta monachus*)	0/4	0
Aquatic birds	Anatidae	Common Shelduck (*Tadorna tadorna*)	0/1	0
		Brent Goose (*Branta bernicla)*	0/2	0
		Eurasian Teal (*Anas crecca)*	0/3	0
		Mute Swan (*Cygnus olor)*	0/4	0
		Mallard (*Anas platyrhynchos)*	0/1	0
		TOTAL	52/209	24.88

**Table 4 animals-12-02889-t004:** Wild bird groups associated with *Campylobacter* spp.

Category	N° pos/ N° Tested	%	*p*-Value	OR	CI_95%_
Diurnal birds of prey	6/53	11.32	*<* 0.001	Reference	
Nocturnal birds of prey	22/60	36.67		4.54	1.67–12.31
Strictly or prevalently insectivorous birds	8/50	16.00		1.49	0.48–4.65
Omnivorous birds	13/18	72.22		20.37	5.35–77.50
Prevalently piscivorous birds	3/13	23.07		2.35	0.5–11.02
Granivorous and frugivorous birds	0/4	0		NA	NA
Aquatic birds	0/11	0		NA	NA

Dependent variable is *Campylobacter* spp. positive/negative status. OR: Odds ratio, CI_95%_: 95% Confidence Interval, NA: Not applicable due to zero positive samples. Reference group is diurnal birds of prey.

**Table 5 animals-12-02889-t005:** Detection of *Campylobacter* in wild birds in external or internal aviaries.

	Indoor Aviary		Outdoor Aviary	
Species	N° pos/N° Tested	%	N° pos/N° Tested	%
Long-eared Owl *(Asio otus)*	3/4	75.00	11/27	40.74
Magpie *(Pica pica)*	3/6	50.00	3/4	75.00
Lesser Kestrel (*Falco naumanni*)	0/2	0	5/28	17.85
Common Gull (*Chroicocephalus ridibundus*)	1/1	100	1/3	33.33
Mistle Thrush (*Turdus viscivorus*)	1/1	100	0/1	0
TOTAL	8/14	57.14	20/63	31.74
*p*-value	0.074			
OR	2.87		Reference	
CI_95%_	0.88–9.37			

Dependent variable is *Campylobacter* spp. positive/negative status. OR: Odds ratio, CI_95%_: 95% Confidence Interval. Reference group is outdoor aviary.

**Table 6 animals-12-02889-t006:** Detection of *Campylobacter* according to the age of birds.

	Nestling/Young		Adult	
Species	N° pos/N° Tested	%	N° pos/N° Tested	%
Long-eared Owl (*Asio otus*)	11/25	44.00	3/6	50.00
Common Kestrel (*Falco tinnunculus*)	3/7	42.86	3/16	18.75
Total	14/32	43.75	6/22	27.27
*p*-value	0.218			
OR	2.07		Reference	
CI_95%_	0.64–6.68			

Dependent variable is *Campylobacter* spp. positive/negative status. OR: Odds ratio, CI_95%_: 95% Confidence Interval. Reference group is adult.

**Table 7 animals-12-02889-t007:** Average body temperature of wild birds and *Campylobacter* detection.

Species	Average Body Temperature (°C)[Variance]	Average Body Temperature of Positive Birds (°C)	Average Body Temperature of Negative Birds (°C)	N° Pos/N° Tested	Positivity Mean (%)	*p*-Value	OR	CI_95%_
Little Owl (*Athene noctua)*	41.5 [0.23]	41.2	41.9	6/17	35.29	0.003	2.73	0.68–10.87
Long-eared Owl (*Asio otus*)	41.8 [0.24]	41.6	42.1	14/31	45.16		4.12	1.25–13.57
Barn Owl (*Tyto alba*)	39.4 [0.22]	-	39.4	0/7	0		NA	
Lesser Kestrel (*Falco naumanni*)	40.7 [0.57]	40.7	40.8	5/30	16.66		Reference	
Common Buzzard (*Buteo buteo*)	42.5 [1.26]	-	42.5	0/17	0		NA	
Other large diurnal birds of prey *	42.2 [1.0]	-	42.2	0/8	0		NA	
Common Kestrel (*Falco tinnunculus*)	41.4 [0.27]	41.1	41.8	6/23	26.09		1.76	0.46–6.72

Dependent variable is *Campylobacter* spp. positive/negative status. OR: Odds ratio, CI_95%_: 95% Confidence Interval, NA: Not applicable due to zero positive samples. The reference group is the Lesser Kestrel. * Other species of large diurnal birds of prey: Eurasian Hobby (*Falco subbuteo*), Eurasian Sparrowhawk (*Accipiter nisus*), Peregrine Falcon (*Falco peregrinus*), Red Kite (*Milvus milvus*).

**Table 8 animals-12-02889-t008:** Antibiotic resistance of *Campylobacter jejuni* and *C. coli* from wild birds.

		N° of Resistant Strains (%)		
	Antibiotics	*C. jejuni* (48)	*C. coli* (3)	Total (51)
Macrolides	Azithromycin	2 (4.2)	1 (33.3)	3 (5.9)
	Erythromycin	1 (2.1)	0 (0)	1 (2)
	Chloramphenicol	0 (0)	0 (0)	0 (0)
Quinolones	Ciprofloxacin	21 (43.7)	2 (66.7)	23 (45.1)
	Enrofloxacin	15 (31.2)	1 (33.3)	16 (31.4)
	Nalidixic Acid	10 (20.8)	2 (66.7)	12 (23.5)
Tetracyclines	Tetracycline	6 (12.5)	3 (100)	9 (17.6)
Aminoglycosides	Gentamicin	0 (0)	0 (0)	0 (0)
Sulphonamides	Trimethoprim/Sulfamethoxazole	25 (52.1)	2 (66.7)	27 (52.9)

**Table 9 animals-12-02889-t009:** Multidrug resistance found in *C. jejuni* and *C. coli* strains detected from wild birds.

N° of Drugs	Antibiotics	*C. jejuni* (48)	*C. coli* (3)	Total (51)
2	CIP/ENR	1 (2.08)	0 (0)	1 (1.96)
	CIP/SXT	3 (6.25)	0 (0)	3 (5.88)
	NA/SXT	1 (2.08)	0 (0)	1 (1.96)
	TE/SXT	1 (2.08)	0 (0)	1 (1.96)
	Sub-total	6 (12.5)	0 (0)	6 (11.76)
3	CIP/ENR/NA	2 (4.17)	0 (0)	2 (3.92)
	CIP/ENR/SXT	6 (12.5)	0 (0)	6 (11.76)
	CIP/NA/SXT	2 (4.17)	0 (0)	2 (3.92)
	AZM/E/TE	1 (2.08)	0 (0)	1 (1.96)
	Sub-total	11 (22.91)	0	11 (21.56)
4	CIP/NA/TE/SXT	1 (2.08)	1 (3,33)	2 (3.92)
	CIP/ENR/NA/SXT	2 (4.17)	0 (0)	2 (3.92)
	AZM/CIP/ENR/SXT	1 (2.08)	0 (0)	1 (1.96)
	Sub-total	4 (8.33)	1 (3.33)	5 (9.8)
5	CIP/ENR/NA/TE/SXT	2 (4.17)	0 (0)	2 (3.92)
6	AZM/CIP/ENR/NA/TE/SXT	0 (0)	1 (3.33)	1 (1.96)

AZM: azithromycin; CHL: chloramphenicol; CIP: ciprofloxacin; ENR: enrofloxacin; E: erythromycin; CN: gentamicin; NA: nalidixic acid; TE: tetracycline; SXT: trimethoprim-sulfamethoxazole.

## Data Availability

Data is contained within the article.

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
