# Peer review of "Prevalence and Antimicrobial Resistance of Campylobacter jejuni and Campylobacter coli in Wild Birds from a Wildlife Rescue Centre"

_animals, 2022, doi:10.3390/ani12202889_

Round 1

Reviewer 1 Report

I very much enjoyed reading the research article reporting Campylobacter species in wild birds at a wildlife rescue centre. It is a very topical article, and it adds to the dearth of literature regarding bacterial carriage in wild animals, and the presence and prevalence of AMR.

The study is well planned out, and well executed and well written throughout.

I have a few comments/ questions, but they are only minor.

I wonder if you undersell the results. I agree that they are only from that rescue centre, and that biosecurity should be increased. However, could you say that this is more widespread, and as such, maybe an issue for wild birds, and thus lead to transmission of Campylobacter to other species? Or even to humans?

Line 69- Gram needs capitalising

Line 99- should Salmonella be italicised?

Line 129- were any of these animals treated with antibiotics during their stay at the rescue centre?

Lines 142-152- please italicise Campylobacter

Line 167- often referred to as stained rather than coloured

Line 171- was this all the strains isolated? And if not, how were they chosen?

Lines 245 and 246- molecules may be better as antibiotics?

Table 9- perhaps need to define the antibiotic abbreviations somewhere?

Line 292- perhaps better as bacteria rather than germ?

Line 367- Campylobacter wants to be capitalised and italicised

But overall, a very nice study and well written, so my congratulations to the authors.

Author Response

Dear reviewer 1,

thank you for your comments. The manuscript has been accordingly modified. Below, I reported the replies to your comments. The changes according to your suggestion are coloured in yellow and tracked in the manuscript by “Track changes” function of word system.

Best regards

Elena Circella

Reviewer 1. I very much enjoyed reading the research article reporting Campylobacter species in wild birds at a wildlife rescue centre. It is a very topical article, and it adds to the dearth of literature regarding bacterial carriage in wild animals, and the presence and prevalence of AMR.

The study is well planned out, and well executed and well written throughout.

I have a few comments/ questions, but they are only minor.

Comment. I wonder if you undersell the results. I agree that they are only from that rescue centre, and that biosecurity should be increased. However, could you say that this is more widespread, and as such, maybe an issue for wild birds, and thus lead to transmission of Campylobacter to other species? Or even to humans?

Reply. Thank you for your comment. The reviewer raises an important point. The transmission of infection from wild birds to human has never been documented, to our knowledge. The direct transmission from infected dogs to children has been reported. However, wild animals are considered as potential carriers for the bacteria for zootechnical species.

Comment. Line 69- Gram needs capitalising

Reply. It has been corrected

Comment. Line 99- should Salmonella be italicised?

Reply. It has been italicised

Comment. Line 129- were any of these animals treated with antibiotics during their stay at the rescue centre?

Reply. Few birds which suffered for traumatic lesions were previously treated with antibiotics but only individuals that stopped the treatments at least one month before the collection of samples were included in the investigation

Comment. Lines 142-152- please italicise Campylobacter

Reply. It has been italicised

Comment. Line 167- often referred to as stained rather than coloured

Reply. The word “coloured” has been changed in “stained”

Comment. Line 171- was this all the strains isolated? And if not, how were they chosen?

Reply. Fifty-one out of 52 isolated strains were tested for antimicrobial sensitivity because one strain has grown with difficulty on the agar

Comment. Lines 245 and 246- molecules may be better as antibiotics?

Reply. “Antibiotics” has been replaced by “molecules”

Comment. Table 9- perhaps need to define the antibiotic abbreviations somewhere?

Reply. They are defined in Material and methods section (paragraph 2.3) and now are reported also under the table 9.

Comment. Line 292- perhaps better as bacteria rather than germ?

Reply. “germ” has been replaced by “bacteria”

Comment. Line 367- Campylobacter wants to be capitalised and italicised

Reply. The name Campylobacter has been corrected

Comment. But overall, a very nice study and well written, so my congratulations to the authors.

Reply. Thank you very much

Reviewer 2 Report

Dear Authors,

Thank you for submitting this interesting paper that investigates the prevalence of different Campylobacter species in a bird rescue centre. The paper has a clear human and avian health background.

At current however, there seem to be some large revisions required in the manuscript to ensure the work is scientifically robust. I have attached the PDF version of the manuscript with specific comments. Additionally, please address the following points:

1. Citations. Please ensure that all points that are not common knowledge are cited. There are many points raised in this manuscript which are not evidenced with citations.

2. Groupings. The birds in the study seem to be grouped based on diet or based on activity periods. This is unrepeatable as it isn't clear what sources were used to identify diets, let alone why some species were categorised based on their day or night activity. Please reformat so this is repeatable - use avian Families instead.

3. Background of the birds. The background to the birds (e.g. time spent in the rescue centre, outside or inside, reason for entering the centre) is likely to affect the prevalence/absence of Campylobacter. Please ensure this is considered in your statistical testing. A binomal logistic regression would allow you to do this.

4. Scientific names. Ensure that scientific names are consistently included when mentioning a species for the first time in text. They should be in italics. 

Author Response

Dear reviewer 2,

thank you for your comments. The manuscript has been modified accordingly. Other references were added, and the list has been renumbered. All changes that you required are coloured in green in the manuscript.

Below, I reported the replies to your comments.

Best regards

Elena Circella

Dear Authors,

Thank you for submitting this interesting paper that investigates the prevalence of different Campylobacter species in a bird rescue centre. The paper has a clear human and avian health background.

At current however, there seem to be some large revisions required in the manuscript to ensure the work is scientifically robust.

Comment. I have attached the PDF version of the manuscript with specific comments.

Reply. We modified the manuscript according to all specific comments reported in your PDF version 

Comment. Additionally, please address the following points:

Citations. Please ensure that all points that are not common knowledge are cited. There are many points raised in this manuscript which are not evidenced with citations.

Reply. References have been added in the text of the manuscript particularly in all points indicated by the reviewer.

Comment. Groupings. The birds in the study seem to be grouped based on diet or based on activity periods. This is unrepeatable as it isn't clear what sources were used to identify diets, let alone why some species were categorised based on their day or night activity. Please reformat so this is repeatable - use avian Families instead.

Reply. Thank you for this suggestion. Tables 1 (in Material and methods) and 3 (in Results) have been modified according to your suggestion about taxonomy. In addition, scientific references concerning diets of birds have been provided in the text of the manuscript (Discussion: lines 319-331)

Comment. Background of the birds. The background to the birds (e.g. time spent in the rescue centre, outside or inside, reason for entering the centre) is likely to affect the prevalence/absence of Campylobacter. Please ensure this is considered in your statistical testing. A binomal logistic regression would allow you to do this.

Reply. Thank you for this suggestion. Indeed, this way of analysing the data according to the time spent in the rescue centre would be very interesting. Unfortunately, however, this kind of information, except for some individuals, is not available in the current dataset. We might consider carrying out the suggested type of analyses in a future project. Instead, the analyses based on housing (outdoor/external aviaries or indoor/internal aviaries) was possible only for some species as reported in table 5.

In this study, birds entered in the centre only for debilitation, dehydration, traumatic lesions or young unweaned individuals were only included. The information has been more clearly reported in the text (lines 132-133). Animals affected by infectious diseases were not tested. Therefore, the reason for the selected animals entering the centre has minor relevance for prevalence of Campylobacter, in our opinion.

Comment. Scientific names. Ensure that scientific names are consistently included when mentioning a species for the first time in text. They should be in italics.

Reply. All scientific names have been reported in italics after mentioning the species for the first time, according to your suggestion.

Round 2

Reviewer 2 Report

Dear Authors,

Many thanks for submitting this revised version of the manuscript for review. You have taken into account the feedback provided on the initial review of the paper. You have also shown clearly where changes have been made to the work, as shown with the highlighted sections of text. The reformatting using species taxonomy rather than purported diet is a more scientifically appropriate method. In light of the revisions, the paper is now in a much better position for consideration.